# Universal Functional Regression with Neural Operator Flows

**Yaozhong Shi**
California Institute of Technology
yshi5@caltech.edu

**Angela F. Gao**
California Institute of Technology
afgao@caltech.edu

**Zachary E. Ross**
California Institute of Technology
zross@caltech.edu

**Kamyar Azizzadenesheli**
NVIDIA Corporation
kamyara@nvidia.com

## Abstract

Regression on function spaces is typically limited to models with Gaussian process priors. We introduce the notion of *universal functional regression*, in which we aim to learn a prior distribution over non-Gaussian function spaces that remains mathematically tractable for functional regression. To do this, we develop Neural Operator Flows (OPFLOW), an infinite-dimensional extension of normalizing flows. OPFLOW is an invertible operator that maps the data space into a Gaussian process, allowing for exact likelihood estimation of point evaluations of functions. OPFLOW enables robust and accurate uncertainty quantification via drawing posterior samples. We empirically study the performance of OPFLOW on regression tasks with data generated from Gaussian processes with known closed-form posterior distribution as well as highly non-Gaussian real-world earthquake time-series with an unknown closed-form posterior distribution.

## 1 Introduction

Inference on function spaces is essential to the physical sciences and engineering. It is often desirable to infer the function given a sparse number of observations. There are numerous problems in which functional regression plays an important role, such as inverse problems, forecasting, and data imputation/assimilation. However, regression on function spaces can be particularly challenging in real world problems where the underlying stochastic process is often unknown.

Much of the work on functional regression and inference has relied on Gaussian processes (GPs) [18]. GP regression (GPR) provides several advantages for function space inference including robustness and mathematical tractability. Despite widespread adoption, the assumption of a GP prior for functional inference problems can be rather limiting, particularly in scenarios where the data is heavy-tailed or distributions are multimodal. This underscores the need for models with greater expressiveness, allowing for regression on data arising from unknown stochastic processes. We refer to such regression problems as universal functional regression (UFR).

To solve UFR problems, we believe it requires a model with two primary components. First, the model needs to be capable of learning priors over data that lies in function spaces. There has been much recent progress on learning priors on function spaces [17, 13, 19, 7, 16, 1, 6, 8, 5]. Second, the models need a framework for performing functional regression with learned function space priors and likelihoods–a component that is critically missing from the previous works described above. Addressing these challenges not only expands the models available for functional regression but also enhances our capacity to extract meaningful insights from complex datasets.

Workshop on Bayesian Decision-making and Uncertainty, 38th Conference on Neural Information Processing Systems (NeurIPS 2024).

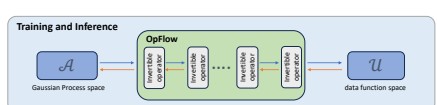
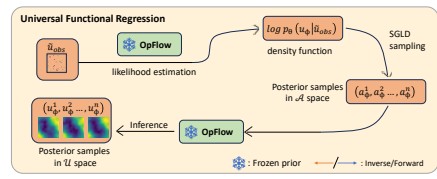

Figure 1: OPFLOW is composed of invertible operators. For the universal function regression task (UFR), OPFLOW is the learned prior which provides likelihoods for function point evaluation. $\tilde{u}_{obs}$ is the noisy observations, $u_\phi$ is the posterior function, and $u_\phi = \mathcal{G}_\theta(a_\phi)$, where $\mathcal{G}_\theta$ is the learned forward operator.

We aim to address this gap by developing a viable formulation for UFR with a novel learnable bijective operator, OPFLOW, that extends normalizing flows to function spaces. OPFLOW consists of an invertible neural operator trained to map the data point-process distribution to a known and easy-to-sample GP distribution. Given point value samples of a data function, OPFLOW allows for computing the likelihood of the observed values, a principled property held a priori by GPs. Using this property, we formally define the problem setting for UFR and develop an approach for posterior estimation with Stochastic Gradient Langevin Dynamics (SGLD) [21]. We demonstrate UFR with OPFLOW on various Gaussian and non-Gaussian processes.

## 2 Related work

**Neural Operators.** Unlike traditional neural networks that primarily work with fixed-dimensional vectors, neural operators are a new paradigm in deep learning designed to operate on functions. Neural operators learn maps between infinite-dimensional function spaces making them inherently suitable for a wide range of scientific tasks involving partial differential equations (PDEs) [10, 11, 12].

**Function space generative modeling with Neural Operators.** Several discretization invariant generative models on function spaces have been recently developed using Neural Operators including Generative Adversarial Neural Operator (GANO) [17], Denoising Diffusion models (DDO) [13], and Variational Autoencoding Neural Operator (VANO) [19]. All of these methods have exhibited superior performance over their finite-dimensional counterparts by directly learning mappings between function spaces. However, they lack the ability to evaluate exact likelihoods of samples.

**Normalizing Flows.** Normalizing flows are a class of flow-based finite-dimensional generative models, that are usually composed of a sequence of invertible transformations [9, 4, 3]. By gradually transforming a simple probability distribution into a more complex target distribution using an invertible architecture, normalizing flows enable exact likelihood evaluation and direct sampling. Traditional normalizing flows are defined on finite-dimensional spaces and constrained to evaluating likelihoods on fixed, regular-size grids.

**Nonparametric Bayesian and Gaussian Process Regression.** Nonparametric Bayesian models are defined on an infinite-dimensional parameter space, yet they can be evaluated using only a finite subset of parameter dimensions to explain observed data samples [15]. These models provide a flexible regression framework by offering analytically tractable posteriors. Despite the great advantages of these models, they heavily depend on the chosen prior, which can restrict their adaptability in complex real-world scenarios [2, 20]. Within the family of nonparmateric Bayesian models, GPR is a robust framework characterized by its analytical posteriors [18]. However, GPR assumes both the prior and posterior are Gaussian, which can limit its applicability.

## 3 Neural Operator Flow

We introduce the Neural Operator Flow (OPFLOW), an innovative framework that extends finite-dimensional normalizing flows to infinite-dimensional function spaces. The architecture is shown schematically in Fig. 1. OPFLOW retains the invertible structure of normalizing flows, while directly operating on function spaces. It allows for exact likelihood estimation for point estimated functions. OPFLOW is composed of a sequence of layers, each containing actnorms [9], domain partitioning, and affine coupling. In particular, domain partitioning mimics the checkerboard pattern in [4, 9].

**Training.** The goal of OPFLOW is to learn a mapping between samples from the data space $\mathcal{U}$ to the latent space $\mathcal{A}$ where $a \in \mathcal{A}, a : \mathcal{D}_{\mathcal{A}} \to \mathbb{R}^{d_{\mathcal{A}}}$ and $u \in \mathcal{U}, u : \mathcal{D}_{\mathcal{U}} \to \mathbb{R}^{d_{\mathcal{U}}}$. Let $\mathbb{P}_{\mathcal{U}}$ and $\mathbb{P}_{\mathcal{A}}$ as probability measures defined on $\mathcal{U}$ and $\mathcal{A}$, respectively. Since OPFLOW is a bijective operator, we only need to learn the inverse mapping $\mathcal{F}_\theta : \mathcal{U} \to \mathcal{A}$, as the forward mapping is immediately available.

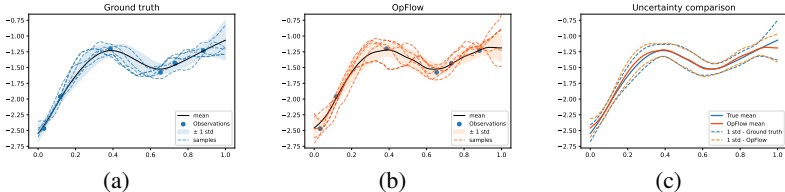

Figure 2: OPFLOW regression on GP data. (a) Ground truth GP regression with observed data and predicted samples (b) OpFlow regression with observed data and predicted samples. (c) Uncertainty comparison between true GP and OpFlow predictions.

We train OPFLOW in a similar way to other infinite-dimensional generative models [17, 19, 13] with a special addition of domain alignment. The space of functions for $\mathcal{A}$ is taken to be drawn from a GP, which allows for exact likelihood estimation and efficient training of OPFLOW by minimizing the negative log-likelihood. Since OPFLOW is discretization agnostic [17, 10], it can be trained on various discretizations of $\mathcal{D}_{\mathcal{U}}$ and can be later applied to a new set of discretizations. In the limit, it can be applied to the whole of $\mathcal{D}_{\mathcal{U}}$ [10]. We now define the training objective $\mathcal{L}$ for OPFLOW training as follows,

$$\mathcal{L} = -\min_{\theta \in \Theta} \mathbb{E}_{u \sim \mathbb{P}_{\mathcal{U}}}[\log p_\theta(u|_D)], \log p_\theta(u|_D) = \log p(a|_{D_{\mathcal{A}}}) + \sum_{i=1}^{s} \log |\det(\frac{\partial(v^i|_{D_{\mathcal{V}}^i})}{\partial(v^{i-1}|_{D_{\mathcal{V}}^{i-1}})})|. \quad (1)$$

The inverse operator is composed of $s$ invertible operator layers with $\mathcal{F}_\theta := \mathcal{F}_\theta^s \circ \mathcal{F}_\theta^{s-1} \cdots \mathcal{F}_\theta^0$, and gradually transforms $v^0$ to $v^1, \cdots v^{s-1}, v^s$, where $v^0 = u, v^s = a$. The alignment naturally holds within the domains associated with $v^0, v^1, \cdots v^s$ as each layer of $\mathcal{F}_\theta$ is an invertible operator. $\mathcal{V}^i$ is the collection of positions for function $v^i$ with $D_{\mathcal{V}}^0 = D, D_{\mathcal{V}}^s = D_{\mathcal{A}}$, and each data point has its own discretization $D$.

**Universal Functional Regression with OPFLOW.** We outline an algorithm for UFR with OPFLOW for general Bayesian inference problems. We assume that OPFLOW has been trained previously and sufficiently approximates the true stochastic process of the data. We thus have an invertible mapping between function spaces $\mathcal{U}$ and $\mathcal{A}$. We use the trained OPFLOW as a learned prior distribution and fix the parameters for functional regression.

Similar to GPR, suppose we are given $\tilde{u}_{obs}$ which consists of pointwise and potentially noisy observations of a function $u$ observed at points in the set $D \subset \mathcal{D}_{\mathcal{U}}$. We refer to the noise-free evaluations at $D$ as $u_{obs}$, with the additive noise $\epsilon \sim \mathcal{N}(0, \sigma^2)$. We aim to infer the function's values on a set of new points $D'$ where $D \subset D'$ given the observation $\tilde{u}_{obs}$. For the likelihood of the values on points in $D'$, i.e., the likelihood of $u_\phi$, we have,

$$\log p_\theta(u_\phi|\tilde{u}_{obs}) = -\frac{\|\tilde{u}_{obs} - u_{\phi|D}\|_2^2}{2\sigma^2} + \log p_\theta(u_\phi) - r\log\sigma - \frac{1}{2}r\log(2\pi) - \log p_\theta(\tilde{u}_{obs}), \quad (2)$$

where $p_\theta(u|_D)$ denotes the probability of $u$ evaluated on $D$ with the learned prior (OPFLOW), and $r$ is the cardinality of the set $D$ with $r = |D|$. Maximizing Eq. 2 results in the maximum *a posteriori* (MAP) estimate $p_\theta(u_\phi|\tilde{u}_{obs})$ where $\overline{u}_\phi$ denotes the MAP estimate given $\tilde{u}_{obs}$. For sampling the posterior in Eq. 2, we utilize the fact that OPFLOW is a bijective operator and the target function $u_\phi$ uniquely defines $a_\phi$. Therefore, drawing posterior samples of $u_\phi$ is equivalent to drawing posterior samples of $a_\phi$ in latent space where we can use SGLD with Gaussian random field perturbation. We utilize SGLD to sample $p_\theta(u_\phi|\tilde{u}_{obs})$. We initialize $a_\phi|\tilde{u}_{obs}$ with the MAP estimate of $p_\theta(u_\phi|\tilde{u}_{obs})$, i.e., $\overline{u}_\phi$, and follow Langevin dynamics in the latent space to sample $u_\phi$.

## 4 Universal Functional Regression, Experiments

In this section, we aim to provide an evaluation of OPFLOW's capabilities for UFR, as well as generation tasks. We consider several datasets, composed of both Gaussian and non-Gaussian processes, as well as a real-world dataset of earthquake recordings with highly non-Gaussian characteristics.

Here, we show UFR experiments using OPFLOW. For training, we take $\mathcal{A} \sim \mathcal{GP}(\mu, k)$ with $k$ being the Matern covariance parameterized by length scale $l$ and roughness $\nu$. The mean function $\mu$ is set to zero over $\mathcal{D}$. The noise level ($\sigma^2$) of observations in Eq. 2 is 0.01 for all regression tasks.

**Gaussian Processes.** This experiment aims to duplicate the results of classical GPR with OPFLOW. For training we use 30,000 GP samples with a Matern covariance parameterized by $l_{\mathcal{U}} = 0.5, \nu_{\mathcal{U}} = 1.5$. For the latent space GP, we use $l_{\mathcal{A}} = 0.1, \nu_{\mathcal{A}} = 0.5$. The resolution of $\mathcal{D}$ is set to 128. Although GPR requires the GP mean and covariance of the data distribution to be known, OPFLOW learns them instead. For regression, we infer the posterior from just 6 random observation points in $\mathcal{D}$. Fig. 2 displays the observed points and analytical solution for GP regression, along with the posterior from OPFLOW. The predicted mean and uncertainty of OPFLOW agree well with the ground truth, demonstrating the ability to accurately capture the posterior distribution.

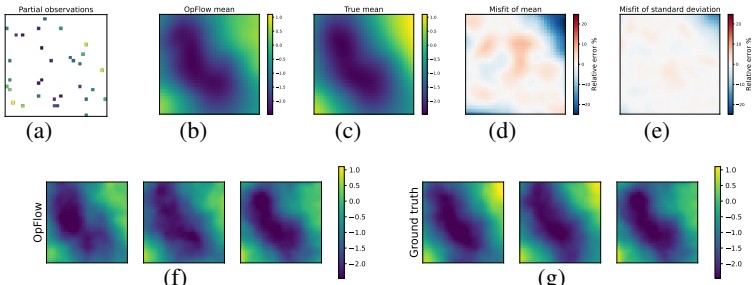

Figure 3: OPFLOW regression on a 32×32 Gaussian random field. (a) 32 random spatial observations. (b) Predicted mean from OPFLOW. (c) True mean from GP regression. (d) Error of the predicted mean. (e) Error of predicted uncertainty. (f) Samples from OPFLOW. (g) Samples from GP regression.

**Gaussian random fields.** We build a 2D dataset of 20,000 32x32 Gaussian random fields with $l_{\mathcal{U}} = 0.5$ and $\nu_{\mathcal{U}} = 1.5$. To train the OPFLOW prior, we use a GP latent space with $l_{\mathcal{A}} = 0.1, \nu_{\mathcal{A}} = 0.5$. In this experiment, the posterior can be computed analytically. In Fig. 3, we perform inference with only 32 random spatial observations. OPFLOW reliably recovers the true posterior mean and variance without assuming that the true data are from a Gaussian Process a priori. However, OPFLOW does have a large amount of error in the upper right corner where there are few observations.

**Seismic waveforms regression.** In this experiment, OPFLOW regression is applied to a real-world non-Gaussian dataset consisting of earthquake ground motion records from Japan [14]. We downsampled the time-series to 10Hz and used $l_{\mathcal{A}} = 0.05, \nu_{\mathcal{U}} = 0.5$. After training, we perform inference on an unseen time series and show results from UFR (Fig. 4). OPFLOW is able to generate samples that match the data and accurately capture key characteristics including various types of seismic waves arriving at different times. The standard GP model struggles in this complex setting.

## 5 Conclusion

We introduced Neural Operator Flows (OPFLOW), an invertible infinite-dimensional generative model that offers a learning-based solution for end-to-end universal functional regression, which has the potential to outperform functional regression with Gaussian processes. Looking ahead, OPFLOW may enable new avenues for analyzing functional data by learning directly from the data.

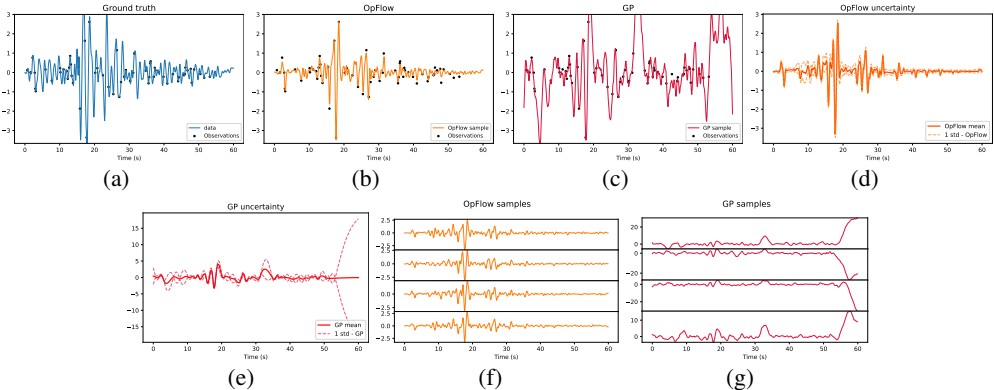

Figure 4: OPFLOW regression on seismic time series data.

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
