# OpenReview forum: "Universal Functional Regression with Neural Operator Flows"
_NeurIPS.cc/2024/Workshop/BDU — NeurIPS BDU Workshop 2024 Poster_

### Official Review · Reviewer_sZYE · 2024-09-25
**A potential contribution to functional regression in machine learning**

**Rating:** 8
**Confidence:** 5

**Review:**

This paper proposes an infinite-dimensional extension of normalizing flows, Neural Operator Flows, that map to Gaussian process base processes. The authors provide a framework for Bayesian functional regression with priors learned via Neural Operator Flows (akin to posterior inference in a Gaussian process in which the prior hyperparameters have been tuned via marginal likelihood). The proposed method relies on MAP estimation and SGLD to sample from the posterior.

The work is important in that flexible functional prior specification and functional inference could improve on current methods for approximate Bayesian inference in machine learning. The paper is mostly clear, but more detail on OpFlow (particularly how domain partitioning works on $\mathbb{R}^d$) would be beneficial. Furthermore, the originality of the paper is not entirely clear; the related work section should probably touch on some related papers using normalizing flows and GPs (e.g.,  https://arxiv.org/abs/2011.01596) and on other functional regression approaches (e.g., functional variational inference, such as https://arxiv.org/abs/1903.05779). The examples presented are somewhat limited, and the seismic example does not in my view contribute much to the paper, as it’s clear that a stationary GP will underperform and it's not clear that OpFlow performs exceedingly well in terms of usual metrics like MSE of the mean prediction or coverage of uncertainty intervals (even though the result sort of “looks like” seismic data).

Some questions for the authors to consider:

1. The presented examples all use Matern kernels for the base GP; how much does the choice of GP in the latent space affect the results?
2. Can you elaborate on connections to other literature involving GPs and normalizing flows, including https://arxiv.org/abs/2011.01596 ?
3. None of the results really show “extrapolation uncertainty”; how does the model perform compared to GPs when extrapolating outside the observed data points?
4. How well does the proposed method achieve coverage via its uncertainty intervals?

---

### Official Review · Reviewer_NTod · 2024-09-26

**Rating:** 4
**Confidence:** 3

**Review:**

Overall, the paper lacks clarity and structure, making it difficult to follow the main contributions. The method is barely explained, and the authors provide only a simple diagram, which is poorly described. Without a clear description of the model, it is difficult to see where Equation 2 comes from; it would be helpful to have a proper explanation of it. Moreover, as far as I understand, the model seems to be a transformed Gaussian Process (GP) with normalizing flows, a concept already explored by Maroña et al. (2021). The authors should have discussed how their model relates to this previous work and highlight any differences.

All the figures were barely readable; I had to zoom in significantly to understand the images.

---

### Decision · Program_Chairs · 2024-10-09

Accept (Poster)